# Unraveling ADAR-Mediated Protein Recoding: A Proteogenomic Exploration in Model Organisms and Human Pathology

**DOI:** 10.3390/ijms26146837

**Published:** 2025-07-16

**Authors:** Viacheslav V. Kudriavskii, Anna A. Kliuchnikova, Anton O. Goncharov, Ekaterina V. Ilgisonis, Sergei A. Moshkovskii

**Affiliations:** 1Lopukhin Federal Research and Clinical Center of Physical-Chemical Medicine of Federal Medical Biological Agency, 119435 Moscow, Russia; wkudriavskii@gmail.com (V.V.K.); ulteran@gmail.com (A.O.G.); moshrffi@gmail.com (S.A.M.); 2Biochemistry Department, School of Biomedicine, Pirogov Russian National Research Medical University, 117997 Moscow, Russia; 3Institute of Biomedical Chemistry, 119121 Moscow, Russia; a.kliuchnikova@gmail.com

**Keywords:** proteogenomics, RNA editing, RNA-dependent adenosine deaminase, ADAR, shotgun proteomics, targeted proteomics, proteoforms

## Abstract

This paper summarizes the results of multi-year studies performed by our research team, focusing on an analysis of protein recoding mediated by messenger RNA editing by ADAR adenosine deaminases. Searching for ADAR-mediated protein recoding was performed in the central nervous system of the model organisms, fruit fly and mouse, as well as in the human proteomic datasets. The proteogenomic approach has made it possible to identify dozens of editing events in the proteome, thus validating the results of transcriptomic studies. The observed recoding events in animals, ranging from insects to mammals, mainly affect the cytoskeletal components and proteins involved in synaptic transmission. In humans, recoding changes are most often observed in the central nervous system or tumor tissues. Over 15 million editing sites have been identified in humans; only a few thousand of those can potentially yield amino acid substitutions. Using a proteogenomic approach, dozens of protein recoding sites are identified, demonstrating their origin in ADAR RNA editing. Moreover, this revealed that the level of recoding at specific sites is not directly related to the abundance of ADAR enzymes per se or their target proteins. The recoding processes probably have differential regulation of interactions at the mRNA level that is yet to be clarified.

## 1. Introduction. RNA Editing by ADAR Adenosine Deaminases

Along with advances in methods used to detect subtle alterations in protein structure, it has also become possible to search for proteoforms [1]. Like protein isoforms, proteoforms share structural similarities and often perform the same function. However, while isoforms originate from different genes, proteoforms arise from the same gene but exhibit variations due to transcript-level changes (e.g., alternative splicing or RNA editing) or post-translational modifications [2]. In this paper, an attempt is made to describe another type of proteoform made by amino acid substitutions in protein sequences as a result of adenosine-to-inosine (A-to-I) RNA editing (protein recoding) and their biological meaning.

A-to-I RNA editing is a natural post-transcriptional modification process resulting from the activity of adenosine deaminases acting on RNA (ADARs). ADARs bind to the double-stranded RNA (dsRNA) regions and catalyze hydrolytic deamination of adenosine at the C6 position, thus causing conversion of adenosine to inosine [3,4]. In humans and other mammals, the ADAR family comprises three enzymes and is encoded by the *ADAR*, *ADARB1*, and *ADARB2* genes [5]. All three enzymes have a similar domain structure, but only the products of the *ADAR* and *ADARB1* genes (ADAR1 and ADAR2, respectively) exhibit catalytic activity [6]. Meanwhile, they are characterized by different substrate specificities, tissue localizations, and regulatory mechanisms. ADAR1 is expressed in all the tissues and is predominantly involved in editing long, perfectly paired dsRNA regions [7] that often originate from repetitive genomic elements such as Alu repeats or long interspersed nuclear elements (LINEs) [8,9]. An important feature of such dsRNAs is their ability to bind to intracellular dsRNA sensors, similar to viral nucleic acids. If this occurs, a downstream cascade activates the expression of interferon type I (IFN I) and other proinflammatory cytokines, which, in turn, negatively affect cell viability [10,11,12]. In this case, the biological significance of ADAR1 is that the newly formed inosine ceases to pair with uridine because of the absence of an amino group, thus destabilizing the RNA duplex and preventing activation of the interferon response to endogenous dsRNAs. Thus, over 90% of RNA editing events are mediated by ADAR1 and occur specifically in Alu repeats, which tend to form dsRNA regions [9]. ADAR1 deficiency causes erythropoiesis impairment and liver cell destruction in mice, resulting in embryonic lethality by day 12 of development [13]. In humans, hereditary ADAR1 deficiency causes a form of Aicardi–Goutières syndrome. This hereditary autoimmune disease is characterized by increased production of IFN I and central nervous system damage [14].

Furthermore, because of inosine’s structural similarity to guanosine, it forms hydrogen bonds predominantly with cytidine at all stages of information transfer. As a result, if an editing event occurs in the coding region of a mRNA, it can lead to amino acid substitution in the protein [15,16]. This type of RNA editing is known as protein recoding and is mainly mediated by ADAR2. The position of the edited nucleotide within the codon determines whether the edit is recoding or silent (Table 1). Editing events at the first or second position usually lead to nonsynonymous (recoding) changes, since they alter the core codon identity. In contrast, editing the nucleotide at the third (“wobble”) position is often synonymous. However, synonymous substitutions, regardless of their position within a codon, can affect translation by altering codon optimality [17], elongation speed [18], and co-translational folding dynamics [19]. In rare cases, adjacent or multiple editing events can even cause ribosome stalling [20].

As previously noted, ADAR2 serves as the primary mediator of protein recoding, though ADAR1 has also been demonstrated to catalyze specific recoding events, such as CCNI R>G [21] and AZIN1 S>G [22]. Discussions on the nucleotide context determinants of recoding within dsRNA regions predominantly focus on ADAR2 substrate specificity. However, shared factors influencing both ADARs also contribute to protein recoding outcomes. Structural disruptions (e.g., bulges or mismatches) in dsRNA regions enhance editing when located upstream of the target adenosine—specifically within −35 nt for ADAR1 and −26 nt for ADAR2 [23]. In addition, editing efficiency increases with larger mismatches, peaking at 3-nucleotide mismatches (median maximal effect). Unpaired adenosines with opposite cytosine in dsRNA regions exhibit elevated editing rates [23,24,25,26]. While both ADARs rely on a 5′ nucleotide context, ADAR2 uniquely depends on 3′ neighbors [23,27,28]. ADAR1 primarily targets long dsRNA regions, often formed by repetitive elements (e.g., Alu and LINE repeats) [7]. ADAR2 favors structured RNAs with loops or local imperfections [29]. Lastly, editing efficiency is further modulated by regulators that impact ADARs protein abundance [30] and protein–protein interactions [31].

Protein recoding as a result of RNA editing is not widespread compared to all editing events; however, it has interesting consequences: the emergence of new proteoforms and diversification of the entire proteome [32]. This review focuses on this type of RNA editing and methods for exploring it at the protein level.

## 2. Protein Recoding via A-to-I RNA Editing

The first targets of A-to-I recoding were established in mammalian glutamate [33,34] and serotonin receptors [35]. Ionotropic glutamate receptors underlie fast excitatory neurotransmission in vertebrates. They include N-methyl-D-aspartate receptors (NMDA receptors, NMDARs), kainate receptors (KARs), and α-amino-3-hydroxy-5-methyl-4-isoxazolepropionic acid receptors (AMPA receptors, AMPARs). AMPARs consist of four subunits (GluA1–4), GluA1/GluA2 heterotetramers being the most frequent combination in the forebrain. RNA editing sites in AMPA receptors are among the best-studied elements of ADAR2-mediated RNA editing. They were shown to play both mechanistic and functional roles. RNA editing at one of these sites in transcripts of the GRIA2, GRIA3, and GRIA4 genes results in substitution of arginine for glycine (R>G) in the respective subunits: GluA2, GluA3, and GluA4. Editing at this site affects the pattern of pre-mRNA splicing and controls the rate of receptor recovery after desensitization [34]. An even more important recoding event is related to the substitution of glutamine for arginine (Q607R) in the GluA2 subunit. This first detected example of protein recoding in mammals results in a voltage-independent decrease in membrane permeability to calcium ions [36,37]. This site is normally edited in nearly 100% of transcripts, which is critical for survival. Insufficient editing at this site is associated with development of amyotrophic lateral sclerosis [38,39] and some types of malignant gliomas [40]; the absence of editing at this site causes early death in mice [41]. It is the only known editing site associated with such a severe phenotype. In fact, postnatal lethality in ADAR2 knockout mice is reduced when the Q607R substitution in the GluA2 subunit is introduced directly via adenosine-to-guanosine (A-to-G) genome editing. As a result, the GRIA2 transcript initially encodes arginine without any need for RNA editing [41]. ADAR2 knockout mice carrying this substitution in the GRIA2 gene remain viable, yet have moderate neurological and immunological disorders [42].

The serotonin 2C receptor (5-HT2cR), which is also expressed in the central nervous system, is another important target for editing. Its RNA is edited at five different sites leading to three amino acid substitutions (I156, N158, and I160). An important feature of these sites is that they are edited independently of each other and not in all the transcripts. Hence, recoding can potentially give rise up to 24 different proteoforms with different characteristics [35,43]. Recoding of 5-HT2cR was shown to reduce its potency to serotonin and binding to G protein, thereby regulating receptor sensitivity [44,45]. In contrast to the Q607R GluA2 site, mice expressing only the genomically encoded version of the receptor (INI amino acids) were developing normally, whereas those expressing only the fully recoded proteoform (VGV amino acids) exhibited growth retardation, polyphagia, reduced fat mass, and increased energy expenditure, being indicative of hyperactivation of the sympathetic nervous system [46]. Moreover, these sites are differentially edited in response to stress [47], while the disruption of 5-HT2cR RNA editing is associated with neurological disorders such as anxiety, depression, and suicidal tendencies [48].

## 3. Functional Significance of Protein Recoding

A legitimate question that is raised in most reviews focusing on RNA editing concerns the functional meaning of protein recoding in the human proteome. There are several models attempting to explain it. The first theory suggests that most recoding sites are either neutral or moderately deleterious [49], which is supported by reports that elevated RNA editing is typical of tumors [50,51,52,53]. However, most tumors are characterized by reduced RNA editing of the sites of ADAR2, an enzyme that mostly modifies exons [54,55,56,57]. According to the second and third hypotheses, A-to-I RNA editing either protects against deleterious mutations with guanosine-to-adenosine substitution [58] or, vice versa, precedes the advantageous adenosine-to-guanosine transition in the genome [59,60]. Furthermore, an emerging hypothesis suggests that ADAR-mediated protein recoding may expand the functional proteome in organisms with constrained genome sizes. This phenomenon could significantly influence evolutionary adaptation, as the RNA editing machinery itself is heritable while the specific editing events remain plastic. This creates a unique mechanism for generating phenotypic diversity without permanent genomic changes [61,62].

Post-transcriptional and post-translational processes make up the major contribution to the complexity of higher organisms [63,64]. These mechanisms can diversify the proteome depending on time, conditions, and cell type, thus causing functional heterogeneity between tissues, developmental stages, brain regions, or even between cells within the same tissue [65]. In particular, protein recoding by A-to-I RNA editing facilitates proteome diversification [66,67]. All the copies of an allele of a gene, or to be more specific, a particular splice variant, encode the same protein in all cells of an organism under all the conditions. However, recoding can give rise to a plethora of proteins from a single genome-encoded variant, thus providing the organism with new adaptation tools [68]. Since RNA editing can be fine-tuned for partially editing the target site, a cell can simultaneously produce different numbers of both edited and unedited proteoforms. As a result, recoding opens up a much broader range of opportunities for transcriptome modification than genomic mutations do. Therefore, RNA editing is a mechanism for transient proteome diversification in response to environmental or physiological signals [69]. Several studies have demonstrated how the levels of RNA editing at certain recoding sites are altered depending on the state of an organism. For example, it has been shown that editing of different transcripts modulates the circadian rhythms of transcripts in the mouse liver [70], and some alterations in RNA editing are related to sleep [71]. Importantly, many studies have demonstrated that editing of GluA2, 5-HT2cR, and many other recoding targets is altered under various pathological conditions [72]. Furthermore, recoding of the potassium channel in octopus correlates with ambient temperature, although it still remains unclear whether this effect is caused by rapid acclimatization or long-term adaptation [73]. The adaptive role of A-to-I RNA editing in response to temperature fluctuations has been well characterized in cephalopods, whose neural transcriptomes undergo extensive recoding by ADAR enzymes [74]. In *Octopus bimaculoides*, nearly one-third of the A-to-I RNA editing landscape—comprising over 20,000 sites—is altered within hours of a temperature shift. This dynamic editing fine-tunes the activity of key neural proteins, including synaptotagmin and kinesin-1 [75].

An ability to respond to short-term selective pressure is an additional important advantage that protein recoding has over genomic mutations [76]. Unlike genetic mutations, alterations are transient and are well suited for eliciting an immediate response to external stimuli and accelerating adaptation to variations in the internal or environmental conditions without compromising the genome. For example, the peak of ADAR expression and editing levels during coral spawning leads to the occurrence of more than a thousand recoding events during gamete release but is not observed in adult corals [77]. This remarkable increase in protein diversity may improve gamete adaptability without manipulating DNA. Each of the many gametes being formed has its own recoding variant, and this diversity may increase the chance that at least some of them will survive during the development under dynamic and diverse environmental conditions. However, it is still unclear how significant these protein recoding events are and whether they actually confer any selective advantage.

## 4. Searching for Recoded Proteins Using the Proteogenomic Approach

### 4.1. Methodological Aspects

Detection of protein recoding events in the proteomes is limited to using the conventional proteomic databases, since they rely exclusively on genomic information. Like in the case of cancer proteogenomics, where tumor genomes are used, specialized search databases focusing on specific substitutions need to be created. Because of the degeneracy of the three-letter code, approximately 20% of A-to-I RNA editing events can give a rise up to 19 amino acid substitutions (Table 1). By relying on nonsynonymous substitutions of adenosine found in exons, one can obtain information about amino acid substitutions in proteins in silico and then use it to search for recoded peptides in entire proteomes (Figure 1). The proteogenomics methodological pipeline closely resembles that of conventional proteomics. However, the proteome database is expanded to include modified proteoforms derived from genomic mutations or transcriptomic variations, including RNA editing and alternative splicing. Subsequent database searches against this expanded reference should yield novel findings. Thus, once a proteomic database is modified so as to include recoded peptides, A-to-I RNA editing events can be identified in panoramic proteomics data. The new findings require careful validation using targeted proteomic approaches—such as multiple reaction monitoring (MRM) or parallel reaction monitoring (PRM)—with isotope-labeled peptide standards. Additionally, unlike mutated proteins, proteoforms created by A-to-I RNA editing are present in the cell along with genomically encoded variants and exhibit distinctive functional hallmarks [78], and targeted proteomics allows quantifying the ratio between recoded and genomically encoded peptides. This ratio is a key parameter for comparing RNA editing, especially in the context of pathologies. Furthermore, the results obtained using this approach have particular research significance, since the correlation between RNA expression and an abundance of its protein products is established to be weak [60,61,62]. Nevertheless, researchers usually confine themselves to detection of the levels of RNA editing only at the transcriptomic level because of the technological complexity and high cost of proteogenomic studies followed by validation.

Proteogenomics is mostly implemented in animal studies. Proteogenomics of plants and animals differ primarily due to variations in genome organization (e.g., polyploidy). In plants, the proteogenomic pipeline is primarily employed for protein annotation, functional characterization, and the validation of single-nucleotide polymorphisms (SNPs) and alternative splicing events [79]. While A-to-I RNA editing appears to be absent in plants, it has been documented in fungi [80,81]. However, fungal A-to-I editing occurs via ADAR-independent mechanisms [82,83]. Consequently, the proteogenomic pipeline can also be adapted to verify A-to-I editing events in fungi.

### 4.2. Results of Proteogenomic Searches for ADAR-Mediated Protein Recoding

The aforedescribed approach has been used to study the proteomes of the central nervous system of model organisms: fruit fly [84,85], zebrafish [86], and mouse [87]; for humans, 40 publicly available curated proteomic datasets of various tissues were used [88]. In addition, 311 human proteomic datasets were analyzed in a bulk manner without the filtration of output as an addition to a major transcriptomic study [89]. 

A total of 62 recoding events in 54 proteins have been identified in the central nervous system of the fruit fly [84]. Interestingly, most of these proteins belonged to cytoskeletal components or were associated with synaptic signal transduction, and through the interplay with the SNARE complex in particular. For example, in the latter case, the recoded peptides referred to syntaxin (*Syx1A*), complexin (*cpx*), and calcium-dependent secretion activator (*Cadps*). The level of endophilin A (*EndoA*) recoding was as high as 74% in the proteomes of adult fruit fly, suggesting that its recoding plays a functional role. The affinity between the membrane and the endophilin A BAR domain, the region where the recoding site resides, is ensured by electrostatic interactions between the negatively charged phospholipid heads and the positively charged lysine residues in the concave portion. Thus, the alteration of charge by K>E recoding in endophilin A may change the affinity of protein binding to the membrane, which would be responsible for the regulation of membrane dynamics in neurons. However, this assumption needs to be confirmed by experiments employing recombinant proteins and model membranes.

The recoding sites and levels were detected for 15 developmental stages of the fruit fly using open-access proteomic data [85]. A comparison revealed that recoding begins at a particular stage, although the abundance of the protein per se remains unchanged at all the developmental stages. The cluster of proteins recoded predominantly during the embryonic stage included embryo-specific and cytoskeleton-associated proteins. The larval stages were enriched in the recoded variants of muscle proteins. The largest cluster of recoded proteins, found in late pupae and the imago, was related to neuronal functions.

We have focused on the sites related to synaptic conductance using target analysis of peptides encoded by the *EndoA*, *stol*, *Cadps*, *cpx*, and *Syx1A* genes in the *D. melanogaster* brain at three lifecycle stages: larva, pupa, and the imago. The level of recoding at each site increased as we proceeded from the larval to the imago stage. Hence, the metamorphosis of the fruit fly was accompanied by an increase in the level of recoding, which followed the burst of RNA editing described earlier at the transcriptomic level [90].

Another part of our study focused on searching for recoded peptides in the proteomes of the mammalian, human, and mouse central nervous systems. Using open-access proteomic data, we detected 20 recoding events in 14 proteins in the mouse brain proteome (those shared with humans are presented in Table 2). Notably, the number of detectable sites in both the fruit fly and mouse proteomes was much smaller than expected. For instance, the number of A-to-I protein recoding sites was estimated to be 8000 in the fruit fly and 800–1800 in mammals. Thus, in the fruit fly and mouse proteomes, we observed only 1% and 2–2.7% of predicted recoding events, respectively. This underrepresentation likely stems from technical limitations in detecting low-abundance or poorly ionizing peptides in mass spectrometry. While the number of detected recoded peptides in the zebrafish’s neural tissue proteome was comparatively the same (10 peptides) [86], it accounted for nearly 10% of predicted recoding events, as only 117 recoding sites were identified in the zebrafish transcriptome [91]. Remarkably, seven of those were located within AMPA glutamate receptor subunits, consistent with findings in other species, and three were unique to zebrafish. In contrast, cephalopods, which exhibit high levels of recoding, display particularly high levels of edited peptides: over 10% of all detected peptides in the squid stellate ganglion and giant axon were found to be recoded [66].

In mammals, some of the most intriguing findings emerged from analyzing the relationship between protein abundance and editing levels in mouse brain regions, using open-access proteomic data. We have demonstrated the absence of a correlation between protein abundance and the level of its recoding [88]. For example, the abundance of the GRIA2 glutamate receptor subunit in cerebellar and cortical neurons is approximately the same; however, the level of recoding in the cerebellum was found to be significantly lower compared to that in neurons of the cerebral cortex. Meanwhile, a high level of editing of cytoplasmic FMR1-interacting protein 2, the product of the *CYFIP* gene, was detected in the cerebellum. The second remarkable phenomenon described by us includes alterations in the level of editing with individual’s age, which was investigated in primary neuronal cell cultures of mice. While the abundance of the alpha-filamin (*FLNA*) protein was identical during aging, its recoding increased with age in microglia. On the contrary, the level of recoding of the coatomer subunit α (*COPA*) was higher in young mice. Hence, these results demonstrated that RNA editing is fine-tuned, and the editing level of a particular site is dependent not solely on ADAR abundance. For instance, the stability of secondary mRNA structure influences its level of editing and can be controlled, for example, by RNA splicing [92]. Other intranuclear factors that can interfere with editing and regulate transcripts, which have not been identified yet, are apparently responsible for the selective regulation of recoding processes [87].

Next, having identified 20 editing events in mouse proteomes, we quantitatively verified them in the cerebral cortex and cerebellum of mice used as a model of CNS diseases [88]. Specifically, three types of transgenic mice were utilized: first, Neat1 ncRNA knockout mice modeling a mental disorder reproducing the features of schizophrenia; second, the model of FUS pathology (neurodegeneration used for studying amyotrophic lateral sclerosis); and third, the tauopathy model mimicking Parkinson’s, Pick’s, and Alzheimer’s diseases. While the recoding level of different peptides ranged from 10 to 100% in both brain regions, no significant differences in the recoding of investigated proteins were observed across the diseases. However, we have identified differences in glutamate receptor subunit recoding between the cerebellum and the cerebral cortex, which can serve as a subject for functional research.

Finally, we analyzed 40 open-access human proteomic datasets from the PRIDE repository and detected 33 protein recoding sites; 18 of those were found in at least two datasets. Reanalysis of the proteomes of different tissues allowed us to divide the revealed recoding sites into two groups. The first group comprised recoding sites for ubiquitous proteins (filamins A and B, coatomer subunit α, and HSPA1L chaperone) as well as tumor-specific proteins (Table 2, Figure 2). The second group included the events considered classical and well-studied ones. These included editing targets characteristic of nervous tissue, such as AMPA-type glutamate receptor subunits (GRIA2, GRIA3, and GRIA4) [33]. Among the detected recoded neuronal proteins, there were also the voltage-gated calcium channel gamma-8 subunits (CACNG8 or TARP γ8), functionally related to glutamate receptors [93]; cytoplasmic FMR1-interacting protein 2 (CYFIP2), which is essential for neural development [94]; and the CADPS calcium-dependent secretion activator. Along with brain-specific recoded sites, an R>G recoding event of insulin-like growth factor-binding protein 7 (IGFBP7) was found in the cerebrospinal fluid in all four datasets examined. Based on the structural data for this protein, it was hypothesized that the R>G substitution in the insulin-like growth factor-binding domain could destabilize the interaction between the protein and its ligand [95]. Furthermore, recoding impacts the susceptibility of the IGFBP7 to proteolysis [96]. Similarly, Gabay et al. investigated ADAR-mediated recoding in the human proteome, though their study primarily focused on transcriptomic data [89]. While their transcriptomic analysis was sophisticated, their proteogenomic pipeline lacked refinement, omitting detailed search parameter optimization and result curation in spite of using 311 datasets from the PRIDE repository [97]. Gabay et al. [89] identified 35 recoding events among 138 peptides detected via mass spectrometry, some of them representing sites validated by more rigorous proteomic approaches in our work. At the same time, they did not report some well-characterized recoded sites in their proteomic survey, such as R764G in human glutamate receptor 2/4 (GRIA2/4) and I164V in COPA (Table 2).

Functional significance has already been described for at least two recoding events from the first group. According to the mouse studies, Q>R editing in filamin A is responsible for the regulation of vascular contraction [98]. The second well-known target for RNA editing is coatomer subunit alpha, which is a component of the specific vesicle coating ensuring retrograde protein trafficking from the Golgi complex to the endoplasmic reticulum. Although the I>V substitution at position 164 is conserved, the literature data suggests that increased editing of this site promotes tumor growth, metastatic spread, and invasion; the level of RNA editing correlates with tumor infiltration and unfavorable disease prognosis [99]. A little later, the mechanism underlying these effects was also elucidated [100]. The I64V substitution impairs the ability of COPA to bind to the signaling dilysine motif, thus leading to activation of endoplasmic reticulum stress and, eventually, to apoptosis. In colorectal cancer cells, under disturbed apoptosis conditions, the transcription factors responsible for activating the transcription of apoptosis-initiating genes also upregulated oncogene expression, which stimulated tumor growth, development, and aggressiveness. Remarkably, a completely opposite effect has been demonstrated in another study: unedited COPA was associated with increased expression of PI3K/AKT/mTOR signaling pathway proteins in hepatocellular carcinoma [101]. Additionally, reduced stability of edited I164V COPA was also reported in this study. These seemingly opposing effects suggest that COPA recoding may play divergent roles depending on the cellular context or disease type. Interestingly, both observed molecular mechanisms are not mutually exclusive, since endoplasmic reticulum stress decreases PI3K/AKT/mTOR activity [102]. Hence, edited COPA induces endoplasmic reticulum stress and reduces cell proliferation. The aforementioned studies have explained the functional significance of COPA recoding in the context of alterations in its RNA editing in cancer cells, and the function of recoding at this site in normal cells is subject to discussion and further research. Among other consequences of altered protein recoding is the development of an adaptive immune response against overedited R75G CCNI [21].

The data summarized in Table 2 show that most protein recoding either alters the charge at the edited position or noticeably modifies the structure (Q>R, K>E, R>G, T>A, Y>C). Some variants causing polarity changes (I>M, M>V) are related to insertion or removal of a sulfur atom from the amino acid, as well as removal of the hydroxyl radical (S>G). However, substitutions that do not dramatically alter protein properties and structure, such as I>V and K>R, are of the greatest interest. These events can hardly be explained by existing theories suggesting that editing is a mechanism protecting against deleterious mutations leading to guanosine-to-adenosine substitutions [58] or, conversely, preceding the advantageous adenosine-to-guanosine transition in the genome [59]. However, these substitution variants are in good agreement with the theory that most protein recoding events mediated by adenosine deaminases are either neutral or moderately deleterious [49]. It is also noteworthy that mRNA editing resulting in the I>V substitution was found to be involved in worse progression-free patient survival time in some cancers such as renal carcinoma [99]. Further investigation may reveal that these subtle recoding events could influence protein folding kinetics, alter degradation rates, or modulate post-translational modification accessibility—all of which represent important avenues for future experimental validation.

While beyond the primary scope of our experimental studies, cephalopods—particularly coleoid species including squid, octopus, and cuttlefish—represent exceptionally important organisms for understanding A-to-I RNA editing and protein recoding. Cephalopods exhibit the most extensive RNA editing repertoire known in the animal kingdom, with editing levels that dramatically surpass those observed in other taxa. Their remarkable editing capacity, coupled with complex neural systems and behavioral sophistication, makes them invaluable models for studying the functional consequences of widespread RNA editing. Transcriptome-wide analyses have identified between 54,287 and 86,230 nonsynonymous (recoding) A-to-I editing sites, affecting 6688 to 8537 open reading frames (ORFs) across four coleoid species (*Octopus bimaculoides*, *Octopus vulgaris*, *Doryteuthis pealeii*, and *Sepia officinalis*) [66]. These findings indicate that approximately 65% of coding-sequence editing events lead to amino acid substitutions. Proteogenomic analyses have confirmed that a significant portion of these RNA editing events are translated into protein isoforms. In a targeted mass spectrometry study of the squid giant axon and stellate ganglion, 432 recoding sites were validated at the peptide level [66]. Several of these proteomically validated sites have clear functional implications. In delayed rectifier K+ channels (Kv2), an I>V substitution in the pore region accelerates channel gating and supports rapid nerve conduction in cold-adapted octopuses [73]. This site is nearly 100% edited in polar species but minimally edited in tropical ones, exemplifying temperature-responsive editing. A second edit in the T1 domain of the same channel (R>G) reduces tetramerization efficiency and lowers functional expression, providing a post-transcriptional mechanism to fine-tune excitability [103]. In synaptotagmin-1, an I>V edit in the Ca^2+^-binding C2A domain reduces calcium affinity, likely increasing the neurotransmitter release threshold under cold stress [75]. Similarly, an editing event in kinesin-1 (K>R in the motor domain) modulates axonal transport: the edited form exhibits slower velocity and shorter run length, likely adapting intracellular trafficking to temperature-dependent metabolic constraints [75].

Although conserved RNA recoding events are rare between evolutionarily distant insects, cephalopods, and mammals, similar patterns appear in analogous tissues and functional pathways. In particular, in Drosophila, protein editing predominantly targets presynaptic membrane proteins, especially those regulating neurotransmitter vesicle maturation and release, along with cytoskeletal proteins and ion channels. Coleoid cephalopods also exhibit conserved recoding events linked to synaptic vesicle release [75]. As mentioned above, A-to-I RNA editing is considered a molecular adaptation to ambient temperature changes in poikilotherms, such as fruit flies [104] and cephalopods [74,75]. Insects and cephalopods are distant from each other on the evolutionary tree and, according to manual curation of results, do not share many edited and/or recoded sites in orthologs. However, an I>V recoding in the C2A domain of synaptotagmin in octopus, shown to participate in adapting calcium signaling to cold [75], is also present as an I>M recoding in squids [74], and the same recoding type is found in a *Drosophila melanogaster* ortholog, Syt1 [84]. The latter was demonstrated to exist at the proteome level [84]. Nonetheless, synaptotagmin editing sites were shown to be insensitive to temperature in the original work [104]. Mice and humans share numerous editing sites, many previously identified in biochemical studies—including AMPA-type glutamate receptors, cytoskeletal proteins (e.g., FLNA, CYFIP2), Golgi vesicle formation proteins (alpha-coatomer), and CADPS, which is also edited in Drosophila (though at a distinct site). Zebrafish show recoding in AMPA receptor paralogs and RIMS2B, a synaptic exocytosis regulator [86].

## 5. Conclusions

Protein recoding by ADARs still remains a poorly studied process in molecular biology, especially in humans and other mammals. However, it plays an extremely significant role: from being involved in vital processes in the central nervous system to participation in tumor development. However, although the methods for studying editing directly at the RNA level are elaborated sufficiently well, the effects at the level of proteins as the final products of editing still need to be studied. The proteogenomic approach allows one to identify dozens of protein recoding sites induced by RNA editing enzymes and demonstrate that the level of recoding at specific sites does not directly correlate with abundance of either ADAR enzymes per se or their target proteins. It is likely that recoding processes have differential regulation of the interplay at the mRNA level that has not been characterized yet. Subsequent assessment of detected peptides by targeted proteomics and, in some cases, DIA quantitation [105] will provide more precise data on the recoding level, which can be compared with the results from transcriptomic searching for RNA editing events. The findings obtained in this way are of the greatest research interest, since the correlation between RNA expression and the abundance of its protein products is poor. Thus, quantitation on the protein level harbors a great scope of applications: from narrowing a spectrum of sites for experimental validation, e.g., in neoantigen discovery in cancer [21] and endogenous interventions, including CRISPR-mediated editing, which can generate fully recoded proteoforms, to new findings, which may not be feasible to observe on RNA levels, thus supplying new insight in developmental biology and aging.

A key limitation of existing studies is their reliance on canonical proteomes for database searches, while microproteins translated from short open-reading frames (sORFs) in RNAs that were previously considered non-coding remain underexplored in humans and other organisms [106,107,108]. This gap highlights the need for further proteogenomic investigations. Additionally, current methods for detecting RNA editing events face technical constraints [109], underscoring the necessity for improved detection techniques. Moving forward, we anticipate enhanced rigor in proteogenomic analyses and an expanded search space for novel RNA editing events, achieved through the inclusion of diverse splice isoforms and microproteins in the analytical pipeline.

## Figures and Tables

**Figure 1 ijms-26-06837-f001:**
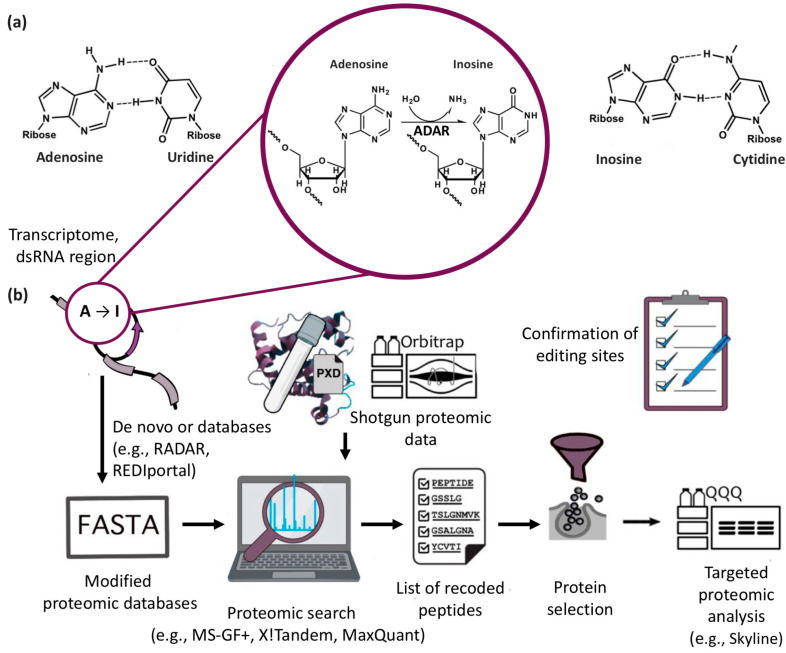
(**a**) ADAR deaminates adenosine and converts it to inosine. Adenosine-to-inosine editing in dsRNA results in replacement of adenine–uracil base pairs with inosine–cytidine base pairs and may cause protein recoding. (**b**) The scheme of the proteogenomic approach, including example tools. The FASTA database extended using the transcriptomic and proteomic data is employed to search for recoded proteins. Peptides of interest are selected and confirmed by targeted proteomic analysis (e.g., PRM, MRM with isotope-labeled peptide standards). The schematic structure and the sequence of steps can be modified for research purposes.

**Figure 2 ijms-26-06837-f002:**
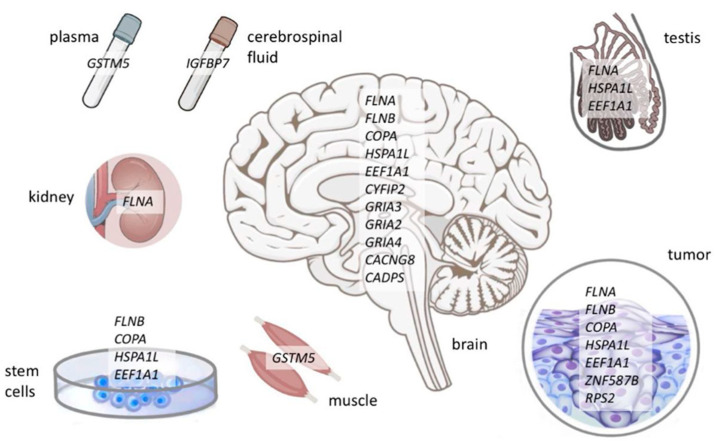
Human protein recoding events by ADARs and their localization.

**Table 1 ijms-26-06837-t001:** Proteogenomics of protein recoding due to RNA editing. Potential amino acid substitutions resulting from the action of ADAR enzymes. Amino acids are color-coded as nonpolar/hydrophobic,
aromatic,
polar/hydrophilic, uncharged, and charged.

Codon	Amino Acid	Edited Codons	Amino Acid After Recoding	Possible Amino Acid Substitutions
UAU UAC	Tyrosine	UGU UGC	Cysteine	Y>C
AAU AAC	Asparagine	AGU AGC	Serine	N>S
AAU AAC	Asparagine	GAU GAC	Aspartic acid	N>D *
CAA CAG	Glutamine	CGA, CGG CGG	Arginine	Q>R
AAU AAC	Asparagine	GGU GGC	Glycine	N>G
AGU AGC	Serine	GGU GGC	Glycine	S>G
ACC ACA ACG ACU	Threonine	GCC GCA, GCG GCG GCU	Alanine	T>A
CAU CAC	Histidine	CGU CGC	Arginine	H>R
AAA AAG	Lysine	GAA, GAG GAG	Glutamic acid	K>E **
AAA AAG	Lysine	AGA, AGG AGG	Arginine	K>R
AAA AAG	Lysine	GGA, GGG GGG	Glycine	K>G
AGA AGG	Arginine	GGA, GGG GGG	Glycine	R>G
GAU GAC	Aspartic acid	GGU GGC	Glycine	D>G
GAA GAG	Glutamic acid	GGA, GGG GGG	Glycine	E>G
AUU AUC AUA	Isoleucine	GUU GUC GUA, GUG	Valine	I>V
AUA	Isoleucine	AUG	Methionine	I>M
AUG	Methionine	GUG	Valine	M>V
UAA UAG UGA	Stop codon	UAG, UGG, UGA UGG UGG	Tryptophan	STOP>W

* The spontaneous deamination of asparagine makes it difficult to reliably identify asparagine-to-aspartic acid substitutions in mass spectrometry analysis; ** causes a charge change.

**Table 2 ijms-26-06837-t002:** The events of mRNA editing by ADAR enzymes detected in mammalian proteomes. Adapted from ref. [88]. The ✓ and ✗ symbols denotes that recoding events have been identified or not, respectively.

Recoded Tryptic Peptide	Protein	Gene	Tissues	Found in Mouse Brain Samples	Identified in [89]: Transcriptome/Proteome
tissue-nonspecific recoding events					
LTVSSL[Q>R]	filamin A	*FLNA*	brain, testis, kidney, pancreas ductal adenocarcinoma cell line, triple-negative breast cancer, B-cell lymphoma	✓	✓/✓
LTV[M>V]SLQESGLK	filamin B	*FLNB*	brain, ovarian cancer cells, meningioma, mesenchymal stem cells	✗	✓/✓
VWD[I>V]SGLR	coatomer subunit alpha	*COPA*	brain, glioma-derived stem cells, ovarian cancer cells, triple-negative breast cancer, mesenchymal stem cells, meningioma	✓	✓/✗
QTQIF[T>A]TYSDNQPGVLIQVYEGER	heat shock 70 kDa protein 1-like	*HSPA1L*	brain, B-cell lymphoma, testis, glioma-derived stem cells, ovarian cancer cells,	✗	✗/✗
QTQIFT[T>A]YSDNQPGVLIQVYEGER			brain, B-cell lymphoma, mesenchymal stem cells, testis	✗	✗/✗
Y[Y>C]VTIIDAPGHR	eukaryotic translation elongation factor 1 alpha 1	*EEF1A1*	brain, glioma-derived stem cells, pancreas ductal adenocarcinoma cell line, ovarian cancer cells	✗	✗/✗
NMI[T>A]GTSQADCAVLIVAAGVGEFEAGISK			glioma-derived stem cells, mesenchymal stem cells, testis	✗	✗/✗
TSLGN[I>M]VK	zinc finger protein 587B	*ZNF587B*	glioma-derived stem cells, brain, ovarian cancer cells	✗	✗/✗
TYS[Y>C]LTPDLWK	40S ribosomal protein S2	*RPS2*	ovarian cancer cells, meningioma	✗	✗/✗
HNLCGETEEE[K>R]	glutathione S-transferase Mu 5	*GSTM5*	muscle, plasma	✗	✓/✗
neural-specific recoding events					
YI[K>E]TSAHYEENK	cytoplasmic FMR1-interacting protein 2	*CYFIP2*	brain	another peptide of this protein was found	✓/✓
GSAL[R>G]NAVNLAVLK	glutamate receptor 3 (GluR-3), flop isoform	*GRIA3*	brain	✓	✓/✓
GSAL[R>G]TPVNLAVLK	glutamate receptor 3 (GluR-3), flip isoform		brain	✓	✓/✓
GSSL[R>G]NAVNLAVLK	glutamate receptors 2 and 4 (GluR2, GluR4), flop isoforms	*GRIA2, GRIA4*	brain	✗	✓/✗
GSSL[R>G]TPVNLAVLK	glutamate receptors 2 and 4 (GluR2, GluR4), flop isoforms		brain	✓	✓/✗
AGGGAGG[S>G]GGSGPSAILR	voltage-dependent calcium channel gamma-8 subunit	*CACNG8*	brain	✗	✓/✗
AGGGRPS[S>G]PSPSVVSEK	calcium-dependent secretion activator 1	*CADPS*	brain	another peptide of this protein was found	✓/✗
GEGEPCGGGGAG[R>G]GYCAPGMECVK	insulin-like growth factor-binding protein 7	*IGFBP7*	cerebrospinal fluid	✓	✗/✗

## Data Availability

All authors confirm that all data and materials support their published claims and comply with field standards and are included in this article.

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
