# Peer review of "Unraveling ADAR-Mediated Protein Recoding: A Proteogenomic Exploration in Model Organisms and Human Pathology"

_ijms, 2025, doi:10.3390/ijms26146837_

Round 1

Reviewer 1 Report

Comments and Suggestions for Authors

This review presents a comprehensive review of ADAR-mediated A-to-I RNA editing events that result in protein recoding, with a focus on their detection through proteogenomic approaches across model organisms and human systems. The topic is timely and relevant, especially as the field increasingly shifts toward multi-omics integration to understand post-transcriptional regulation and proteome diversity.

The authors compile a substantial amount of data and experimental findings, some from their own group, and discuss the biological significance of protein recoding in neural physiology, evolution, and disease contexts. However, while the manuscript covers a broad range of material, several issues regarding organization, language clarity, scientific rigor, and presentation hinder the impact and readability of the review.

Major points:

  1. While Table 2 provides a helpful summary of mammalian ADAR-mediated recoding events, it appears to be largely extracted from a single source (reference [67]) without additional synthesis or comparative commentary. For a review article, it would be beneficial to cross-reference similar findings from other studies. Or discuss what these editing events suggest about tissue specificity, evolutionary conservation, or disease relevance.
  2. The early sections contain overlapping discussions of editing function without clear thematic separation, and the blending of background information with author interpretation reduces clarity. Additionally, the methodological description of proteogenomics is mixed with application data, disrupting the logical flow. A clearer separation between background, interpretation, methods, and results would greatly improve readability.
  3. The manuscript reviews numerous protein recoding events across various species, including humans, mice, and flies. Given the scope of these findings, it would greatly improve readability and clarity if the authors could summarize the key recoding events across species in a single comparative table. This would allow readers to better appreciate the cross-species patterns and functional relevance of ADAR-mediated editing.
  4. Please make a comparison of proteogenomic approach between animal and plant studies, g. DOI: 10.1016/j.tibtech.2023.05.010 etc.
  5. Please expand the Introduction and perspective paragraphs by citing recent publications. Some of them but not limited to are suggested as follows:

1) https://doi.org/10.1007/s44307-023-00007-3

2) …

Minor points:

  1. Figure 1 is useful but not well integrated into the text. Please ensure it is clearly referenced and explained.
  2. What’s the difference between isoforms and proteoforms? The author should give a clear definition of proteoforms when it was first mentioned.
  3. Please correct the format of reference of 44 in page 12.
  4. The place citing Reference [70], can add this citation as well: https://doi.org/10.1007/s44307-024-00010-2 for further describe the RNA structure and neuro-diseases.

Author Response

Comment 1: While Table 2 provides a helpful summary of mammalian ADAR-mediated recoding events, it appears to be largely extracted from a single source (reference [67]) without additional synthesis or comparative commentary. For a review article, it would be beneficial to cross-reference similar findings from other studies. Or discuss what these editing events suggest about tissue specificity, evolutionary conservation, or disease relevance.

Response 1: We have expanded Table 2 to include a comparative analysis between the stringent recoding findings from [67] (now [88]) and the comprehensive transcriptomic/proteomic data reported by Gabay et al. While cephalopods, Drosophila, and mammals share limited conserved recoding events due to their evolutionary divergence, we have supplemented our discussion with a new paragraph at the end of section 4.2 highlighting convergent recoding patterns potentially linked to adaptive processes.

Comment 2: The early sections contain overlapping discussions of editing function without clear thematic separation, and the blending of background information with author interpretation reduces clarity. Additionally, the methodological description of proteogenomics is mixed with application data, disrupting the logical flow. A clearer separation between background, interpretation, methods, and results would greatly improve readability.

Response 2: As our article is a review, we aimed to avoid structuring sections like a traditional research paper. To enhance clarity in the section titled 'Searching for Recoded Proteins Using the Proteogenomic Approach,' we have divided it into two subsections: '4.1. Methodological Aspects' and '4.2. Results of Proteogenomic Searches for ADAR-Mediated Protein Recoding. We hope this will improve the readability and clarity of our work.

Comment 3: The manuscript reviews numerous protein recoding events across various species, including humans, mice, and flies. Given the scope of these findings, it would greatly improve readability and clarity if the authors could summarize the key recoding events across species in a single comparative table. This would allow readers to better appreciate the cross-species patterns and functional relevance of ADAR-mediated editing.

Response 3: In mammals and invertebrates, the editing regions show minimal overlap, meaning a comparative table would essentially comprise two distinct sections. Nevertheless, we included a paragraph at the end of Section 4.2 highlighting shared recoding events and the functional categories of edited proteins.

Comment 4: Please make a comparison of proteogenomic approach between animal and plant studies, g.DOI: 10.1016/j.tibtech.2023.05.010 etc.

Response 4: We have incorporated a new paragraph addressing proteogenomics in plants and fungi (a paragraph at the end of section 3, just after the table 1). However, since A-to-I RNA editing does not occur in plants and our review primarily focuses on animal proteogenomics, we are unable to expand this section further. That said, we have acknowledged the potential application of proteogenomic approaches for studying A-to-I RNA editing events in fungi.

Comment 5: 
Please expand the Introduction and perspective paragraphs by citing recent publications. Some of them but not limited to are suggested as follows:

1) https://doi.org/10.1007/s44307-023-00007-3

2) …

Response 5: We believe that the linked article was added by mistake, as it pertains to thermal proteomics rather than proteogenomics and is not relevant to the manuscript's topic.

Comment 6: Figure 1 is useful but not well integrated into the text. Please ensure it is clearly referenced and explained.

Response 6: We agree. We have reworked figure 1 and reintegrated it into the text with additional description.

Comment 7: What’s the difference between isoforms and proteoforms? The author should give a clear definition of proteoforms when it was first mentioned.

Response 7: We address this important comparison in the introductory section of our paper (p. 3, lines 2-6).

Comment 8: Please correct the format of reference of 44 in page 12.

Response 8: Thanks for noticing this issue. We have corrected the format of this reference.

Comment 9: The place citing Reference [70], can add this citation as well: https://doi.org/10.1007/s44307-024-00010-2 for further describe the RNA structure and neuro-diseases.

Response 9: 

While we acknowledge that the review article you referenced provides valuable insights into RNA structures and their functional consequences, we have chosen to focus specifically on RNA editing regulation in this section. Our discussion centers on how splicing efficiency modulates RNA editing, and we have cited primary literature that directly supports this relationship. We appreciate your thoughtful input and will certainly consider this reference for other relevant sections of our work.

Reviewer 2 Report

Comments and Suggestions for Authors

This review is of  good quality, I have two suggestions for further improvement:

1. Since topic is code-affecting ADARs, you may add a section describing how and why the A->I transition is recognized as G codon by tRNA, perhaps even show a figure explaining the mechanism. Moreover, does the edited position (1-3) in the codon affect translation in any way?

2. Clarify in your review if apart from the double strand formation and stability, there are any other (e.g. editing site sequence neighbours)  RNA sequence features that affect the recoding by ADARs (e.g. the final ratio of edited/non-edited RNA in the cell)

Comments on the Quality of English Language

A few sentences must be corrected, either grammar wrong or content confusing.

Author Response

Comment 1: Since topic is code-affecting ADARs, you may add a section describing how and why the A->I transition is recognized as G codon by tRNA, perhaps even show a figure explaining the mechanism. Moreover, does the edited position (1-3) in the codon affect translation in any way?

Response 1:  Figure 1 has been modified to include the mechanism described in the text (second paragraph of the introduction section). The codon changes can also be seen in Table 1. Regarding the positional effects of edited adenosines within codons (positions 1-3), we have added a new explanatory paragraph in the Results section (page 3, lines 67-78). This addition systematically examines how editing at each codon position impacts translation outcomes.

Comment 2: Clarify in your review if apart from the double strand formation and stability, there are any other (e.g. editing site sequence neighbours)  RNA sequence features that affect the recoding by ADARs (e.g. the final ratio of edited/non-edited RNA in the cell)

Response 2: We have described how the RNA sequence context may influence editing of a site (the neighbor nucleotides, loops, mismatches) with dissecting similar features of both ADARs as well as their differences (p.3, lines 79-93). Furthermore, we have addressed additional regulatory factors known to modulate RNA editing, such as SRSF9 and AMIP2.

Reviewer 3 Report

Comments and Suggestions for Authors

Dozens of protein recoding sites can be found using the proteo-genomic technique, which also shows that these sites emerge following RNA editing by ADARs. Furthermore, data demonstrated that the quantity of ADAR enzymes or their target proteins in general is not directly correlated with the degree of recoding at particular locations. It is yet unclear if the recoding mechanisms have distinct regulation of interactions at the mRNA level. This is a modern and very interesting subject that did originate from both insects and mammals. In this regard, the authors ought to take into account the entire body of current research on the subject, noting not just flies but also the most recent discoveries in the silkworm moth, for example. Peptide editing after RNA editing by ADAR, ribosome peptide mutation, location and diversity of mutations, regulation, which regulatory mechanism could be behind such mutations, impact of peptide mutations on protein gene evolution, etc. are some of the recent findings in insect genetics that were cited in the moth study. All of these elements have been extensively covered in the literature and published after Xuan et al. 2014.

Xuan et al.  2014 Molecular evidence of RNA editing in Bombyx chemosensory protein family. PLoS ONE 9: e86932.

This may lead to a new discussion about the processes that seem to be in place in moths, flies, and mammals. In the review and chapter from Picimbon (2017, 2019), post-ADAR mutations in two chemosensory binding protein families (OBPs and CSPs) are precisely described and located, in accordance with Xuan et al. (2014). Do flies also have similar kinds of mutations? In mammals? Which protein gene families are concerned by the editing mechanisms that the authors are reviewing? 

The majority of the genetic events reported in cephalopods, for example, are not included in the current review, which also lacks a thorough understanding of all various genetic changes in insects. The authors should think about updating their topics of discussion and analyzing the complete current data set for such a wide review prospect. It might aid in the development of some fresh, significant, and pertinent theories regarding the control of the recoding process.

Author Response

Comment 1: 

Dozens of protein recoding sites can be found using the proteo-genomic technique, which also shows that these sites emerge following RNA editing by ADARs. Furthermore, data demonstrated that the quantity of ADAR enzymes or their target proteins in general is not directly correlated with the degree of recoding at particular locations. It is yet unclear if the recoding mechanisms have distinct regulation of interactions at the mRNA level. This is a modern and very interesting subject that did originate from both insects and mammals. In this regard, the authors ought to take into account the entire body of current research on the subject, noting not just flies but also the most recent discoveries in the silkworm moth, for example. Peptide editing after RNA editing by ADAR, ribosome peptide mutation, location and diversity of mutations, regulation, which regulatory mechanism could be behind such mutations, impact of peptide mutations on protein gene evolution, etc. are some of the recent findings in insect genetics that were cited in the moth study. All of these elements have been extensively covered in the literature and published after Xuan et al. 2014.

Xuan et al.  2014 Molecular evidence of RNA editing in Bombyx chemosensory protein family. PLoS ONE 9: e86932.

This may lead to a new discussion about the processes that seem to be in place in moths, flies, and mammals. In the review and chapter from Picimbon (2017, 2019), post-ADAR mutations in two chemosensory binding protein families (OBPs and CSPs) are precisely described and located, in accordance with Xuan et al. (2014). Do flies also have similar kinds of mutations? In mammals? Which protein gene families are concerned by the editing mechanisms that the authors are reviewing?

Response 1:  The A-to-I RNA editing of CSPs represents an excellent example of proteome diversification that was inadvertently overlooked in our initial analysis. We have now included this reference (67) in our updated list. Although evolutionary genomics lies beyond the main scope of our review, we recognize its fundamental importance for understanding RNA editing mechanisms. Since such investigations primarily rely on transcriptomic approaches rather than proteomic/proteogenomic methodologies, we have expanded our discussion to include these perspectives, citing relevant works by Picimbon et al (2017) and Moldovan et al (2022) (section 3, paragraph 1). Furthermore, we have added a dedicated paragraph summarizing conserved recoding patterns across species, which helps clarify the functional protein families where these editing events most frequently occur (the last paragraph at the end of the 4.2 section, p. 15).

Comment 2:  The majority of the genetic events reported in cephalopods, for example, are not included in the current review, which also lacks a thorough understanding of all various genetic changes in insects. The authors should think about updating their topics of discussion and analyzing the complete current data set for such a wide review prospect. It might aid in the development of some fresh, significant, and pertinent theories regarding the control of the recoding process

Response 2: We agree that cephalopods represent a unique and highly informative model system for understanding large-scale RNA editing, particularly in the context of protein recoding. In response, we have expanded Chapter 4 (p.14, lines 408 - 434) to include a more comprehensive overview of recoding events in cephalopods. 

Reviewer 4 Report

Comments and Suggestions for Authors

Dear Authors,

The manuscript entitled "Unraveling ADAR-Mediated Protein Recoding: A Proteogenomic Exploration in Model Organisms and Human Pathology” is a technically rich, highly specialized review, and overall, it provides a solid and accurate synthesis of the literature on ADAR-mediated RNA editing and its impact at the proteomic level. Below you may find my comments.

Recommendations & Clarifications for Scientific Accuracy

  1. Statement on Editing Prevalence and Detection (p. 6–7)

“In the fruit fly and mouse proteomes we observed only 1% and 2–2.7% of predicted recoding events, respectively.”

Clarification Needed: Make it clear that this low detection is due to mass spectrometry sensitivity and peptide detectability limitations, not due to actual absence of editing at the protein level.

Suggested Addition:

"This underrepresentation likely stems from technical limitations in detecting low-abundance or poorly ionizing peptides in mass spectrometry."

  1. Role of ADARs in Cancer (p. 9–11)

“The literature data suggested that increased editing of this site [COPA I164V] promoted tumor growth… while unedited COPA was associated with increased PI3K/AKT/mTOR signaling in HCC.”

Conflict Noted: These interpretations appear contradictory; clarify that the functional consequences of COPA editing may be context- or tissue-dependent.

Suggestion:

“These seemingly opposing effects suggest that COPA recoding may play divergent roles depending on cellular context or disease type.”

  1. Neutrality of I>V or K>R Recoding (p. 12)

“These events can hardly be explained by existing theories...”

Problem: The manuscript rightly questions their biological significance but does not offer potential testable hypotheses.

Suggestion: Propose that such subtle recoding events might impact protein folding kinetics, degradation rates, or PTM accessibility, which remain to be experimentally assessed.

  1. Generalization of Functional Consequences (p. 3–4, 10–11)

“RNA editing is a mechanism for heritable proteome diversification…”

Caution: The word “heritable” might be misleading, as A-to-I editing is not generally inherited in the Mendelian sense.

Rephrase:

“...a mechanism for transient proteome diversification in response to environmental or physiological signals.

  1. Microproteins and sORFs (p. 12–13)

“...microproteins translated from sORFs remain underexplored...”

Suggestion: This is a promising closing remark. Consider strengthening the case by referencing recent studies such as:

van Heesch et al., Cell 2019 on sORFs in the human heart

Chen et al., Science 2020 on translation of lncRNAs

This could reinforce the argument that canonical proteomic pipelines miss potentially edited small proteins.

Additional Suggestions

Table 1: The amino acid substitution table is great but could benefit from color coding or grouping based on physicochemical effects (charge, polarity, size).

Figure 1: Consider labeling proteogenomic steps with example tools (e.g., Skyline for PRM/MRM).

Terminology consistency: Use “editing site” rather than “recoding site” uniformly, except where proteomic-level recoding is explicitly the subject.

Author Response

We sincerely appreciate your thorough review and valuable suggestions for improving our manuscript. We have carefully addressed each of your comments and implemented all recommended revisions. Below we provide point-by-point responses to clarify the changes made:

Comment 2: 

Role of ADARs in Cancer (p. 9–11)

“The literature data suggested that increased editing of this site [COPA I164V] promoted tumor growth… while unedited COPA was associated with increased PI3K/AKT/mTOR signaling in HCC.”

Conflict Noted: These interpretations appear contradictory; clarify that the functional consequences of COPA editing may be context- or tissue-dependent.

Suggestion:

“These seemingly opposing effects suggest that COPA recoding may play divergent roles depending on cellular context or disease type.”

Response 2: We agree that the functional significance of these recoding events has not yet been experimentally validated, and we have highlighted this as an important consideration in our discussion

Comment 4: 

Generalization of Functional Consequences (p. 3–4, 10–11)

“RNA editing is a mechanism for heritable proteome diversification…”

Caution: The word “heritable” might be misleading, as A-to-I editing is not generally inherited in the Mendelian sense.

Rephrase:

“...a mechanism for transient proteome diversification in response to environmental or physiological signals.

Response 4: Indeed, our original wording could have lead to confuse, thus we have rephrase as you suggested

Comment 8: Terminology consistency: Use “editing site” rather than “recoding site” uniformly, except where proteomic-level recoding is explicitly the subject.

Response 8: We have replaced 'recoding site' with 'editing site' where appropriate (e.g., at the beginning of the second chapter)   to ensure consistency in our text. However, in some instances, we retain the term 'recoding'—even when referring to RNA-level modifications in the original studies—because editing at these positions results in protein recoding, which is linked to functional outcomes such as adaptation.

We greatly appreciate your thoughtful review and have incorporated all of your suggestions into the revised manuscript.

Round 2

Reviewer 3 Report

Comments and Suggestions for Authors

This creates a unique mechanism for generating phenotypic diversity without permanent genomic changes.  [61,62].

Remove the dots after changes

Double check the text for typo editing

  1. Picimbon, J.-F. A new view of genetic mutations. amj 2017, 10, doi:10.21767/AMJ.2017.3096.

........ AMJ 2017;10(8):701-715

  see for example........AMPA-type glutamate receptors, cytoskeletal proteins (e.g., FLNA, CYFIP2), Golgi vesicle formation proteins (alpha-coatomer), and CADPS, which is also edited in Drosophila (though at a distinct site). Zebrafish show recoding in AMPA receptor paralogs and RIMS2B, a synaptic exocytosis regulator [86].......

Also CRISPR....etc

Please enlarge the abbreviations list

Author Response

Comment 1: 

This creates a unique mechanism for generating phenotypic diversity without permanent genomic changes.  [61,62].

Remove the dots after changes

Double check the text for typo editing

  1. Picimbon, J.-F. A new view of genetic mutations. amj 2017, 10, doi:10.21767/AMJ.2017.3096.

........ AMJ 2017;10(8):701-715

  see for example........AMPA-type glutamate receptors, cytoskeletal proteins (e.g., FLNA, CYFIP2), Golgi vesicle formation proteins (alpha-coatomer), and CADPS, which is also edited in Drosophila (though at a distinct site). Zebrafish show recoding in AMPA receptor paralogs and RIMS2B, a synaptic exocytosis regulator [86].......

Also CRISPR....etc

Please enlarge the abbreviations list

Response 1: Thank you for your suggestions. We have removed an extra dot and double-checked for any remaining typos. As recommended, we updated reference â„–61 to include *"AMJ 2017;10(8):701-715"*, adjusting the punctuation by replacing semicolons with commas to comply with our style guide. Additionally, we have expanded the abbreviation list and have replaced 'edits' with 'editing events' (p. 3, lines 73 - 79) to improve consistency within our text 

Reviewer 4 Report

Comments and Suggestions for Authors

Dear Authors,

You have successfully performed the required revisions. No more comments from my side. 

Author Response

comment 1: You have successfully performed the required revisions. No more comments from my side. 

response 1: Thank you once again for your valuable contribution to our review! We truly appreciate your suggestions --  they have made the text more accurate and  clear.